# The effect of repeat feeding on dengue virus transmission potential in *Wolbachia*-infected *Aedes aegypti* following extended egg quiescence

**Meng-Jia Lau**[1,2], **Andrés R. Valdez**[2,3], **Matthew J. Jones**[1,2], **Igor Aranson**[2,3], **Ary A. Hoffmann**[4], **Elizabeth A. McGraw**[1,2]*

1 Department of Biology, The Pennsylvania State University, University Park, Pennsylvania, United States of America, 2 The Huck Institutes of the Life Sciences, The Pennsylvania State University, University Park, Pennsylvania, United States of America, 3 Biomedical Engineering, The Pennsylvania State University, University Park, Pennsylvania, United States of America, 4 Pest and Environmental Adaptation Research Group, Bio21 Institute and The School of Biosciences, University of Melbourne, Parkville, Victoria, Australia

* eam7@psu.edu

## Abstract

As *Wolbachia pipientis* is more widely being released into field populations of *Aedes aegypti* for disease control, the ability to select the appropriate strain for differing environments is increasingly important. A previous study revealed that longer-term quiescence in the egg phase reduced the fertility of mosquitoes, especially those harboring the *w*AlbB *Wolbachia* strain. This infertility was also associated with a greater biting rate. Here, we attempt to quantify the effect of this heightened biting behavior on the transmission potential of the dengue virus using a combination of assays for fitness, probing behavior, and vector competence, allowing repeat feeding, and incorporate these effects in a model of $R_0$. We show that *Wolbachia*-infected infertile mosquitoes are more interested in feeding almost immediately after an initial blood meal relative to wild type and *Wolbachia*-infected fertile mosquitoes and that these differences continue for up to 8 days over the period we measured. As a result, the infertile *Wolbachia* mosquitoes have higher virus prevalence and loads than *Wolbachia*-fertile mosquitoes. We saw limited evidence of *Wolbachia*-mediated blocking in the disseminated tissue (legs) in terms of prevalence but did see reduced viral loads. Using a previously published estimate of the extrinsic incubation period, we demonstrate that the effect of repeat feeding/infertility is insufficient to overcome the effects of *Wolbachia*-mediated blocking on $R_0$. These estimates are very conservative, however, and we posit that future studies should empirically measure EIP under a repeat feeding model. Our findings echo previous work where periods of extensive egg quiescence affected the reproductive success of *Wolbachia*-infected *Ae. aegypti*. Additionally, we show that increased biting behavior in association with this infertility in *Wolbachia*-infected mosquitoes may drive greater vector competence. These relationships require further exploration, given their ability to affect the success of field releases of *Wolbachia* for human disease reduction in drier climates where longer egg quiescence periods are expected.

**Data Availability Statement:** All data are available via figshare DOI https://doi.org/10.6084/m9.figshare.25417552.

**Funding:** This study was funded by a grant to EAM (R56 AI155573) from the National Institutes of Allergy and Infectious Diseases (https://www.niaid.nih.gov/). The sponsors/funders played no role in the study design, data collection and analysis, decision to publish or preparation of the manuscript.

**Competing interests:** The authors have declared that no competing interests exist.

## Author summary

*Wolbachia pipientis* is a naturally occurring, maternally inherited insect endosymbiont that was artificially introduced into the mosquito, *Aedes aegypti*, the dominant vector of dengue, Zika, chikungunya, and Yellow Fever viruses globally. This bacterium is being released into mosquito populations in the field for disease control because *Wolbachia* reduces the ability of these co-infecting viruses to replicate as well as spread easily through populations due to a form of reproductive manipulation. Previous research has shown that mosquito eggs experiencing long periods of storage make *Wolbachia*-infected adults more likely to be infertile and exhibit increased biting frequency. Using a combination of behavioral, vector competence, and fitness assays as well as modeling, we show that the increase in biting behavior leads to higher virus prevalence and load in the mosquito. Together, the decreased fertility and the increased virus transmission potential may decrease the efficacy of *Wolbachia*, particularly in dry climates where mosquitoes likely spend longer in the egg stage before hatching. We highlight a range of future experiments needed to fully ascertain the effect of egg quiescence on virus transmission.

## Introduction

*Wolbachia*, a maternally inherited endosymbiotic bacterium widespread in arthropods, reduces the replication of co-infecting viruses in its insect hosts [1,2]. *Wolbachia* is not naturally found in the global mosquito vector, *Aedes aegypti*, but several strains have been transinfected into the mosquito from multiple donor insect species [1,3], forming stable infections. *Wolbachia* limits the replication of the arboviruses dengue, chikungunya [2], Zika [4], and Yellow Fever [5] inside the vector, reducing the potential for transmission. It also limits the reproductive success of *Wolbachia*-free females in populations via the action of cytoplasmic incompatibility (CI), leaving *Wolbachia*-infected females to populate a greater proportion of the next generation, assisting with symbiont spread [6,7]. *Wolbachia*-mediated 'pathogen blocking,' in combination with its self-spreading abilities, form the basis of a global strategy to use the symbiont to limit the incidence of human arboviral diseases. The release of *Wolbachia* into native *Ae. aegypti* populations have led to its spread and the replacement of local mosquito populations in numerous field trials [8–11]. High rates of *Wolbachia* infection in mosquitoes post-release have then been associated with substantial reductions in dengue fever incidence in humans [10,12].

The long-term success of these *Wolbachia*-based 'population replacement' strategies depends on at least four factors: the successful vertical transmission of *Wolbachia*, the strong expression of CI, limited fitness costs associated with *Wolbachia* infection, and the ongoing induction of *Wolbachia*-mediated pathogen blocking [13]. The strength of CI expression is known to be correlated with symbiont density (reviewed in [14]). High ambient temperatures can reduce *Wolbachia* densities, causing weakening of CI [15] and maternal transmission failure [16]. Fitness costs to hosts infected with *Wolbachia*, while often mild, can be greater in the presence of high temperatures [17]. High-temperature conditions have been proposed to explain the poor spread and maintenance of *Wolbachia* in at least one release site [18]. The factors that will affect expression levels of pathogen blocking are less clear. Mosquito genetic background may matter [19], as well as *Wolbachia* strain x mosquito interactions [20]. In the releases in Brazil, for example, the efficacy of *Wolbachia* varied across release zones and did not entirely correlate with the frequency of *Wolbachia* in the population, suggesting an

interaction with local environmental or genetic factors [11,21]. Thus, the success of the 'population replacement' strategies seems to depend on the local context.

Even though ~ten different *Wolbachia* strains have been established in *Ae. aegypti* from a range of donor insect species [22], only variants of two strains have been successfully released and established in the field in replacement strategies: *w*Mel [10,21] and *w*AlbB [8]. The *w*AlbB strain, originally from *Aedes albopictus* [3], shows better resistance to both higher and lower temperatures [23,24] compared to *w*Mel, originally from *Drosophila melanogaster* [1]. Subsequent laboratory investigation found, however, that females infected with *w*AlbB can be infertile as adults and lack matured ovaries if they spend an extended period in the egg stage [25]. Infertility rates can reach 75% in females hatched from eggs stored in a humidity-controlled environment for 11 weeks. This reduction suggests that field releases of this strain may encounter difficulty in regions with distinct dry seasons preventing continuous hatching of eggs. Interestingly, the resulting infertile females also demonstrated an increased feeding frequency, with the potential to drive greater transmission of viruses when the rainy season does arrive [26].

The degree of virus transmission is highly dependent upon mosquito population size, survival, feeding frequency, infection rate, and the extrinsic incubation period (EIP) [27]. *Wolbachia*-mediated pathogen blocking results in a reduction in the infection rate and the viral load [2] and lengthens the EIP [28]. Increased feeding due to egg storage may act counter to pathogen blocking. Studies of *Wolbachia's* effect on virus transmissibility in the laboratory are frequently carried out on mosquito lines with little appreciable egg storage time and usually after a single blood feed, where the mosquitoes feed to repletion [1,2]. In the field, egg quiescence may be common in areas with dry seasons, and successive feeding within a single gonotrophic cycle is common, likely due to the consumption of smaller blood meals on live hosts [29,30]. In this study, we examine how *Wolbachia*-associated infertility and increased blood-feeding post-egg quiescence affect DENV transmissibility. We also examine key life history traits, including longevity and probing behavior. With vector competence and fitness measures, we are then able to model the effect of infertility and increased feeding rate on transmissibility.

## Materials and methods

### Mosquito rearing

Two mosquito lines, a *Wolbachia w*AlbB-infected and wildtype (*Wolbachia*-free), were used in this experiment. The *w*AlbB *Wolbachia* strain originated from *Aedes albopictus* and was previously microinjected into *Ae. aegypti* [3]. Just prior to these experiments, *w*AlbB-infected *Ae. aegypti* females were backcrossed for 3 generations to wild-type males collected from Mérida, Mexico, in 2018. This Mérida line also served as the wildtype, *Wolbachia*-free control. All mosquitoes were maintained in the laboratory under 26 ± 1˚C and 68 ± 5% relative humidity, with a 12:12 hour light: dark cycle. A previous study compared the effect of egg storage (quiescence) for 3, 6, and 11 weeks and showed maximal reductions in fertility and fecundity of *Wolbachia*-infected *Ae. aegypti* [25] at 11 weeks. We, therefore, stored all mosquito eggs (+/- *Wolbachia*) for 11 weeks prior to hatching under the environmental conditions defined above.

Mosquito larvae were reared at a density of 100 larvae/L of water and fed Tropical Fish Food (Tetramin: Tetra Werke, W. Germany). Adults were housed in 18 × 18" square breeding cages (BioQuip) in populations of 300 individuals and fed with 10% sucrose solution *ad libitum*. Fresh human blood from anonymous donors (BioIVT: Westbury, NY) was used for all blood-feeding using a Hemotek membrane feeder (Hemotek Ltd., UK). Post-blood feeding, mosquitoes were immobilized by chilling to sort them into smaller experimental containers.

## Life history traits

We measured the longevity and the probing frequency of *Wolbachia*-infected or uninfected female mosquitoes hatched from quiescent eggs as they are key determinants of mosquito virus transmission at the population level. At 5 ± 1 days post-emergence, mosquitoes were blood-fed (no virus), and then engorged females were relocated into 32 oz plastic containers with mesh lids for experimental groups and replicates. As per a previous report, adult females infected with *Wolbachia*, having experienced extended egg quiescence, exhibited high rates of infertility as caused by undeveloped ovaries (see images within [26]). To confirm that our *Wolbachia*-infected lines (all stored as eggs) also demonstrated high rates of infertility compared to *Wolbachia*-uninfected controls, we dissected 100 *Wolbachia w*AlbB-infected and 20 *Wolbachia* uninfected female mosquitoes and calculated the rates of undeveloped ovaries for each. For longevity, each line was represented by five replicate containers, each containing 30–40 individuals. Mosquito survival was checked every two days from 0 to 62 days post-feeding. During this period, mosquitoes were provided with 10% sucrose and 50% larval rearing water using 2 cotton balls, respectively each changed every 2 days. Once per week, mosquitoes were offered a blood meal for 15 minutes using the Hemotek through mesh lids.

For the probing frequency experiment, each line was represented by 7 replicate containers, each containing 30–40 individuals. Over seven days post the first blood meal, warmed blood was provided once per day (24 ± 1 hours). A probing event was described as follows: an individual landed on the feeder and showed probing or probing attempts for at least 3 seconds. Multiple separate feeding events could be scored for the same mosquito if it left the feeder and then returned for probing during the ten-minute assay window. All mosquitoes/treatments were fed within 2 hours, and the order of feeding each day was randomized. A subset of containers was observed and scored in real time by the experimenter. Given the scale of the design, the remainder were scored after watching videos. Mosquitoes were deprived of both sucrose and water one hour before the start of feeding until the end of the trial.

## Dengue virus infection and quantification

We used the DENV-2 (ET-300; GenBank: EF440433.1) strain, cultured in C6/36 *Ae. albopictus* cells to test the transmissibility of mosquitoes after hatching from quiescent eggs. Cells were grown at 25°C in RPMI 1640 medium (Life Technologies, Carlsbad, CA, USA) with 10% sterile fetal bovine serum (FBS, Life Technologies) and 20 mM HEPES (Sigma-Aldrich, St. Louis, MO, USA). For infection, cells were allowed to grow to 80% confluency, then the medium was replaced with fresh RPMI with 2% FBS, and the virus was inoculated at an MOI of 0.1. Viral concentration of the supernatant was quantified 7 days post-infection via squash buffer extraction (10 mM Tris (pH 8.2), 1 mM EDTA, and 50 mM NaCl) and DENV-G qPCR [31]. The virus was diluted in RPMI medium to a concentration of $1 \times 10^7$ copies/ml and then mixed 1:1 with blood in a flask for mosquito feeding. This concentration was chosen according to a previous study, where ~half of the mosquitoes were expected to be infected 12 days post-infection [32]. The concentration of virus in blood was also quantified, 30 μL blood was pipetted into 270 μL TRI Reagent (Sigma-Aldrich, Cat. no. T9424) and extracted using a Direct-zol RNA 96 Magbead Zymo kit (Zymo Research) on a MagMAX Express 96 system (Applied bio- systems), then we qPCR to quantify the copy of dengue virus using a LightCycler 480 instrument (Roche) following the methods described previously [31]. The viral load in blood for feeding each day was determined by averaging the results from two technical replicates from the same flask.

*Wolbachia* infected or wildtype lines hatched from eggs that had been stored for 11 weeks were separated into two groups at 5 ± 1 day post-emergence: one was fed with dengue

infectious blood once; the other was fed and then provided with an infectious blood meal once every subsequent day for the following 7 days. After the initial infectious blood meal, mosquitoes were immobilized on ice, and engorged individuals were sorted into 16 oz containers for further collection. Each container held ~ 30 mosquitoes. Each infectious feed lasted for 30–40 minutes, and cotton balls soaked with sucrose and water were provided during the feeding. Mosquito leg samples were collected at 8, 11, and 14 days post the initial infectious (dpi) feeding for virus quantification. We chose to study legs for safety and ease in a large-scale design (dissection post-freezing) and because a recent study demonstrated the high efficacy of leg tissue loads in predicting transmission potential with an animal model [33]. A biological replicate was repeated with mosquitoes from different containers under the same experimental conditions. We tested the prevalence and viral load in legs to indicate virus dissemination [33], using 24 individuals per line. At the point of dissection, legs were stored in 300 µL of TRI Reagent and stored at -80°C for virus quantification via high throughput RNA extraction methods. Bodies from the *Wolbachia* infected line were also dissected for *Wolbachia* load determination via DNA extraction. Before legs collections, female fertility was first determined under a microscope (Carl Zeiss Microscopy GmbH, Göttingen, Germany) by assessing the existence of ovaries [26]. This allowed us to score whether the individual leg sample was from a fertile or infertile female. All viral loads were then assessed across 6 treatment groups. All *Wolbachia* uninfected mosquitoes were fertile, and so there were only 2 treatment classes- 'fed once' or 'fed multiple times'. *Wolbachia* infected lines, in contrast, exhibited mixed infertility, and so in addition to 'fed once' and 'fed multiple times,' there was also an additional classification of 'fertile' or 'infertile' (4 total classes). Samples were homogenized using a Bead Ruptor Elite, and then RNA was extracted using a Direct-zol RNA 96 Magbead Zymo kit (Zymo Research, Irvine, CA) on a MagMAX Express 96 system (Applied bio- systems) according to the manufacturer's instructions. Dengue virus copy number was then determined in a Light-Cycler 480 instrument following methods described previously [31]. Samples with a Ct value > 32 were considered as negative because at late Ct the standard curve no longer exhibited a linear relationship. After qPCR, DENV infection prevalence in legs was analyzed using a generalized linear model following a Wald test in R studio software. A log10 transformation was applied to viral load data prior to ANOVA.

## *Wolbachia* quantification

We quantified *Wolbachia* densities in bodies (minus legs) after the single feeding. Bodies were chosen to include the numerous tissues [34] where blocking is likely to occur (midgut, fat body, etc.). In brief, after dissection, bodies were stored in 300 µL of TRI Reagent, and a Direct-zol DNA/RNA Miniprep kit (Zymo Research, Cat No. R2080) was used to extract DNA, followed by a qPCR method to test the relative densities of *Wolbachia* using primers for a *Wolbachia* gene and a host control gene as described previously [1,2]. *Wolbachia* densities were also log-transformed and compared using a Student's t-test.

## Statistical analysis

Data analysis and visualization were performed in R studio software (2022.07.1). We used a log-rank survival test for longevity data through package 'survival' [35] and 'survminer' [36]. For the probing behavior data, viral loads, and prevalence, a generalized linear model for Gaussian distribution was used to compare the effect of *Wolbachia* and the day of feeding, with "emmeans" package used to perform post hoc analysis. The Mann-Whitney U test was used for pair-wise comparisons between *Wolbachia*-infected and wild-type mosquitoes on each day.

## Modeling

Following Smith *et al*. [27] we evaluated a Ross and Macdonald index, where $R_0$ estimates the number of new dengue-infected hosts linked to a single mosquito. To avoid human and external factor dependencies, we re-write the $R_0$ as:

$$R_0 = \frac{a^2 c}{g} exp(-gv),\tag{1}$$

The parameters are as follows; *g* is the time-dependent death rate, obtained from the survival rate (Fig 1), *a* is the biting rate (Fig 2), *c* represents the viral infection rate or prevalence (Fig 3), and *v* represents the Extrinsic Incubation Period (EIP). Given our multiple feeding design, we could not directly estimate viral prevalence at early time points; we, therefore, selected the earliest days of DENV arrival in the saliva from a previously published model for '*Wolbachia* uninfected' as 5 days and 7 days for both classes of *Wolbachia*-infected mosquitoes based on *w*Mel and DENV serotype 2 data [37]. Holding EIP constant across infertile and fertile groups, and for both the single and repeat-feeding groups was a very conservative choice. We did attempt to fit a sigmoidal curve to our prevalence data to estimate EIP but found this unreliable without time points before 8 days. To introduce variation in all measures, we perturbed values by 25%. We used the ChaosPy library [38] to perform this task, 10000 samples were generated, retrieving 10000 model evaluations.

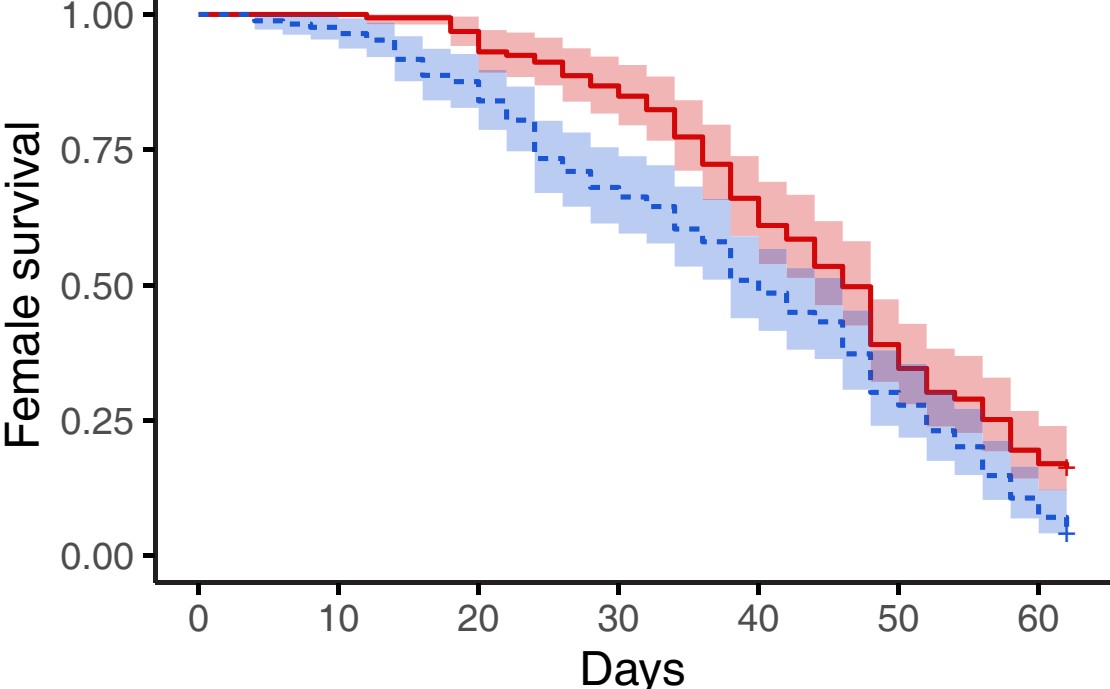

**Fig 1. Longevity of *Wolbachia*-infected and uninfected *Ae. aegypti* females hatched from eggs that were stored for 11 weeks.**
Five replicates, each with 30–40 mosquitoes, were measured for each group. Data are for females that became fully engorged after feeding 5 ± 1 days post-emergence. Females were followed for up to 66 days post-feeding. The shaded areas represent the 95% confidence intervals.

## Results

### Life history traits

Life history traits can affect mosquito vectorial capacity. Here we tested the longevity and probing frequency between the *Wolbachia w*AlbB-infected line and the wildtype line after their eggs were quiescent for 11 weeks. Infertility was the dominant feature of the *Wolbachia* infected line post-egg storage, with 81% of females (n = 100) scoring as infertile based on dissection. In contrast, 0% of wild type females (n = 100) were infertile post egg storage. *Wolbachia*-infected mosquitoes also died faster than wildtype (Fig 1. $\chi2$ = 12.8, df = 1, p < 0.001). The average survival for *Wolbachia*-infected females was 40 days versus 46 days for wild type. We then examined probing frequencies for fully engorged females after their first blood meal over a 7-day time course (Fig 2). The feeding frequency was significantly higher for the *Wolbachia*-infected line compared to the wildtype (GLM: family = Gaussian, t(108) = 2.07, p = 0.04). There was also a significant effect on the day of feeding (GLM: family = Gaussian, t (108) = 5.08, p < 0.001). The *Wolbachia*-infected line showed a significantly higher rate of probing on days 2, 3, 4, and 5 (Mann-Whitney U tests: day 1: Z = -1.00, p = 0.32; day 2: Z = -3.40, p < 0.001; day 3: Z = - 3.38, p < 0.001; day 4: Z = -2.26, p = 0.02; day 5: Z = -2.21, p = 0.03; day 6: Z = -1.63, p = 0.10; day 7: Z = -1.10, p = 0.27). In these first few days, the wild type line shows little interest in probing whereas the rates by the *Wolbachia*-infected line are ~10-fold greater. In later dpi, as probing interest rises in the wildtype line, the *Wolbachia*-infected still feed at roughly twice the rate.

**Fig 2. Probing frequency of *Wolbachia*-infected and uninfected *Ae. aegypti* females hatched from eggs that were stored for 11 weeks.** Seven replicates, each with 30–40 mosquitoes, were measured for each group. Data are for females that became fully engorged after feeding 5 ± 1 days post-emergence. The curves are fitted using a LOESS method and formula = 'y ~ x'. The shaded areas represent the 95% confidence intervals.

## Dengue virus prevalence and load

All mosquitoes were fed with blood mixed with DENV-2 virus at 5 ± 1 days post-emergence, and after this initial infectious blood meal, they were divided into two groups: 'blood-fed once' and 'blood-fed repeatedly.' Females from the latter were provided with an infectious blood meal every day from 1 to 7 dpi. Viral prevalence and load were quantified in leg tissues at 3 time points post-feeding. Legs were chosen as they have recently been described as a better proxy for transmission in wild type mosquitoes [33]. Before RNA extraction, the bodies of *Wolbachia*-infected females were dissected under a microscope and used to score individuals as either fertile or infertile based on ovary appearance. For uninfected mosquitoes, there are 2 classes (fed once or multiply). For *Wolbachia*-infected, there are four (fed once or multiply by x fertile vs. infertile). Because 'experimental replicate' was significant in the model for both infection prevalence ($\chi2 = 25.04$, df = 1, p < 0.001), and viral load ($F_{1,375} = 29.71$, p < 0.001), we analyzed the two replicate experiments separately.

Not surprisingly, infection prevalence (Figs 3 and S1A) rose with time in both replicate experiments (replicate 1: $\chi2 = 16.97$, df = 2, p < 0.001; replicate 2: $\chi2 = 38.01$, df = 2, p < 0.001). Similarly, the multiple feeding events also increased prevalence in both replicates (replicate 1: $\chi2 = 4.65$, df = 1, p = 0.03; replicate 2: $\chi2 = 16.20$, df = 1, p < 0.001). Replicate 1 prevalence was much higher than replicate 2, with several reaching 100% (Figs 3 and S1A). Our data were surprising in that the wild type line either showed lower prevalence in the leg tissue (replicate 1: $\chi2 = 10.43$, df = 2, p = 0.005) or there was no difference compared to *Wolbachia*-infected (replicate 2: $\chi2 = 1.63$, df = 2, p = 0.44), suggesting that *Wolbachia* did not provide clear or consistent blocking at the level of dissemination to the legs. When we provided pairwise comparisons for *Wolbachia*-infected fertile and infertile females, infertile females showed higher infection prevalence than fertile females upon multiple feeding at 8 dpi (post hoc upon GLM: replicate 1: z = 2.197, p = 0.028), but not for 11 or 14 dpi or any of the time points in replicate 2 (all p > 0.05).

The viral load data, in contrast, did reveal strong evidence of blocking in the *Wolbachia*-infected across all dpi. In replicate 1 (Fig 3), 'dpi' ($F_{2,215} = 16.21$, p < 0.001), feeding frequency ($F_{1,215} = 24.88$, p < 0.001), and 'mosquito line' ($F_{2,215} = 8.08$, p < 0.001) were significant. Whereas in replicate 2 (S1B Fig), only the 'mosquito line' ($F_{2,160} = 8.97$, p < 0.001) exhibited a strong effect. The interactions between 'dpi' and the 'feeding frequency' (replicate 1: $F_{2,215} = 3.54$, p = 0.031; replicate 2: $F_{2,160} = 7.96$, p < 0.001) or 'mosquito line' (replicate 1: $F_{4,160} = 3.02$, p = 0.018; replicate 2: $F_{4,160} = 2.49$, p = 0.045) were also significant. On average, across treatments, DENV loads in *Wolbachia*-infected lines were $4.44 \times 10^4$ (infertile) and $4.84 \times 10^4$ (fertile), compared to uninfected mosquitoes where the copy number was $1.45 \times 10^5$. When the *Wolbachia*-infected infertile females were blood-fed repeatedly, the viral load increased ~3 fold to $1.33 \times 10^5$. There was only a two-fold increase for the other two lines between single and repeat feeding to $1.06 \times 10^5$ (*Wolbachia*-infected fertile) and $2.14 \times 10^5$ (wildtype).

Last, we tested for *Wolbachia* density after a single DENV-2 feed (S2 Fig), and all mosquitoes from the *Wolbachia*-infected line were infected. We saw no large difference in *Wolbachia* loads between infertile and fertile females, although there was a trend with the former being higher ($F_{1,68} = 3.76$, p = 0.057).

## Modeling

As described above, we used published estimates of EIP (5 days for wild type, 7 days for *Wolbachia* infected). These are conservative estimates, as EIP may be shorter with repeated feeding, especially for the *Wolbachia*-infertile mosquitoes given their probing behavior in the first few days (Fig 2). Our estimates of survival also contained only two classes (wild type, *Wolbachia*).

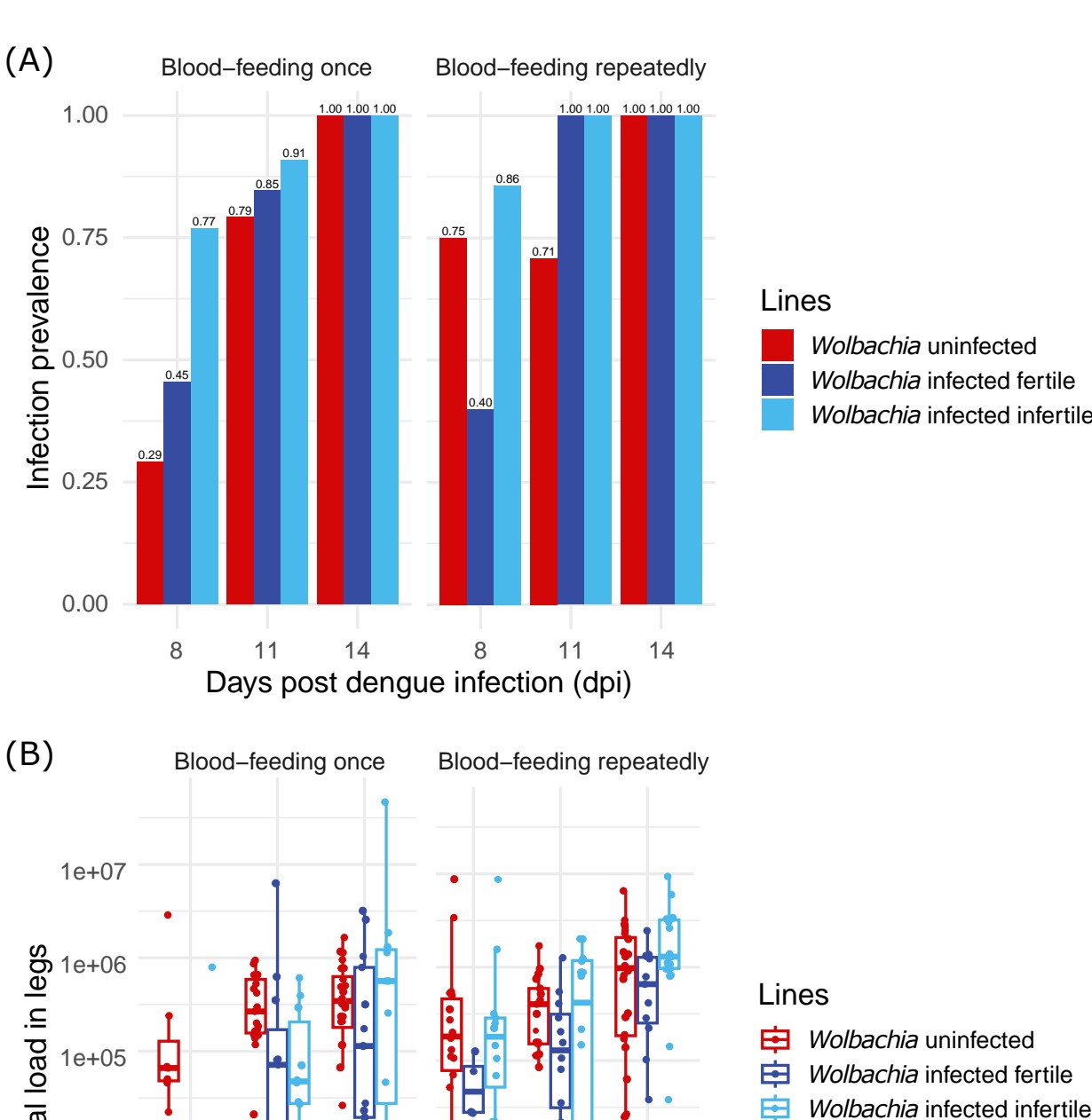

**Fig 3. DENV-2 viral infection following repeat feeding after 11 weeks of egg storage for Replicate 1.** (A) infection prevalence and (B) load in adult female *Aedes aegypti* legs at 8, 11, and 14 days post-infection (dpi). Females were provided with infectious blood every day from 1 to 7 dpi (post the initial infectious blood meal at 0 dpi). Each collection point x treatment is represented by 24 individuals. The fertility status of *Wolbachia*-infected females was identified through ovarian dissection.

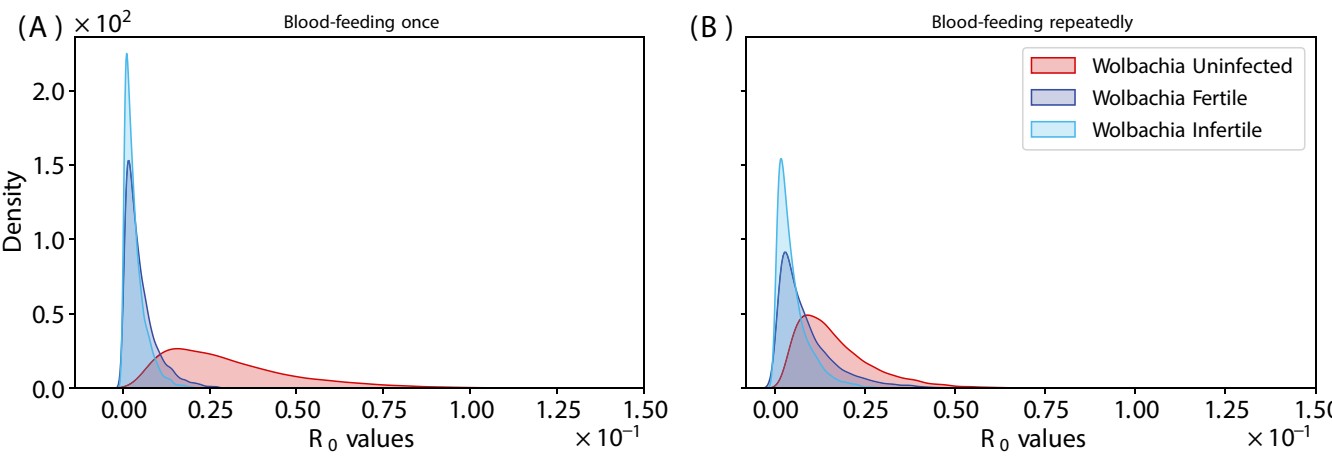

**Fig 4.** $R_0$ density distribution for single (A) and repeatedly blood-fed (B) experiments.

Given the predominance of *Wolbachia*-infected mosquitoes were infertile (~80%), and that previous work has shown that *Wolbachia*-fertile mosquitoes do not feed more frequently than wild type [26], we used the latter estimate for the *Wolbachia*-fertile biting rate. Because the distributions for $R_0$ are non-symmetrical (Fig 4), metrics like the mode and the 90% confidence interval region, rather than the mean, are more useful in comparing across treatments (Table 1). After a single feeding, as expected the *Wolbachia*-free line exhibited the largest $R_0$ (Fig 4A and Table 1) compared to the *Wolbachia*-infected lines. Only for the *Wolbachia*-infected lines does variation in $R_0$, increase with repeat feeding (Fig 4 and Table 1). The opposite is true for the wild type line. The effect of repeat feeding is not large enough to overcome the effects of blocking as expressed as differences in EIP, ie, the wild type line still exhibits the greater $R_0$ although there is a greater overlap of the three curves (less difference). While biting rate, death rate, and EIP were significant determinants of $R_0$ for wild type (Fig 5) and *Wolbachia*-fertile mosquitoes (Fig 6), only the death rate and EIP mattered for *Wolbachia*-infertile mosquitoes (Fig 7). We expected greater differences to emerge between the *Wolbachia* fertile and infertile mosquitoes with multiple feeds (Fig 4), but this is likely due to the use of the same estimates for EIP and death rate for the two *Wolbachia* classes.

**Table 1. Statistical parameters for the $R_0$ probability density distributions depicted in Fig 4 with EIP estimates of 5 and 7 days, respectively for '*Wolbachia*' uninfected and '*Wolbachia*'-infected.** Highlighted in green is the population that transmits most effectively.

| Blood feeding once | | | | | |
|---|---|---|---|---|---|
| Population | Mean | Mode | Median | 5th percentile | 95th percentile | St. Dev |
| *Wolbachia-* | 0.031 | 0.0031 | 0.026 | 0.0082 | 0.070 | 0.019 |
| *Wolbachia+* fertile | 0.0051 | 0.00020 | 0.0037 | 0.00069 | 0.015 | 0.0047 |
| *Wolbachia+* infertile | 0.0035 | 0.00014 | 0.0025 | 0.00048 | 0.0099 | 0.0032 |
| Blood feeding repeatedly | | | | | |
| Population | Mean | Mode | Median | 5th percentile | 95th percentile | |
| *Wolbachia-* | 0.017 | 0.0019 | 0.014 | 0.0046 | 0.038 | 0.0101 |
| *Wolbachia+* fertile | 0.0089 | 0.00036 | 0.0063 | 0.0012 | 0.026 | 0.0081 |
| *Wolbachia+* infertile | 0.0052 | 0.00021 | 0.0037 | 0.00069 | 0.015 | 0.0047 |

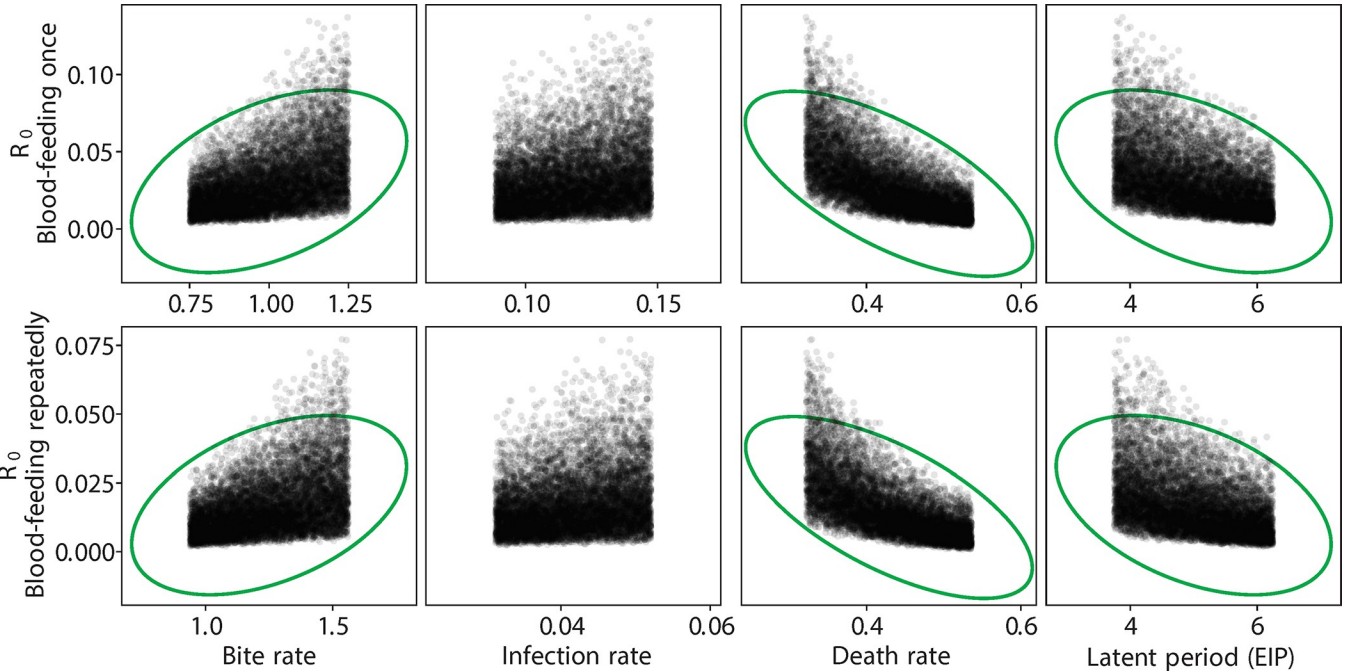

**Fig 5. $R_0$ values vs. experimental parameters for different blood-feeding experiments considering *Wolbachia* uninfected mosquitoes.** Green circles indicate the parameters contributing significantly.

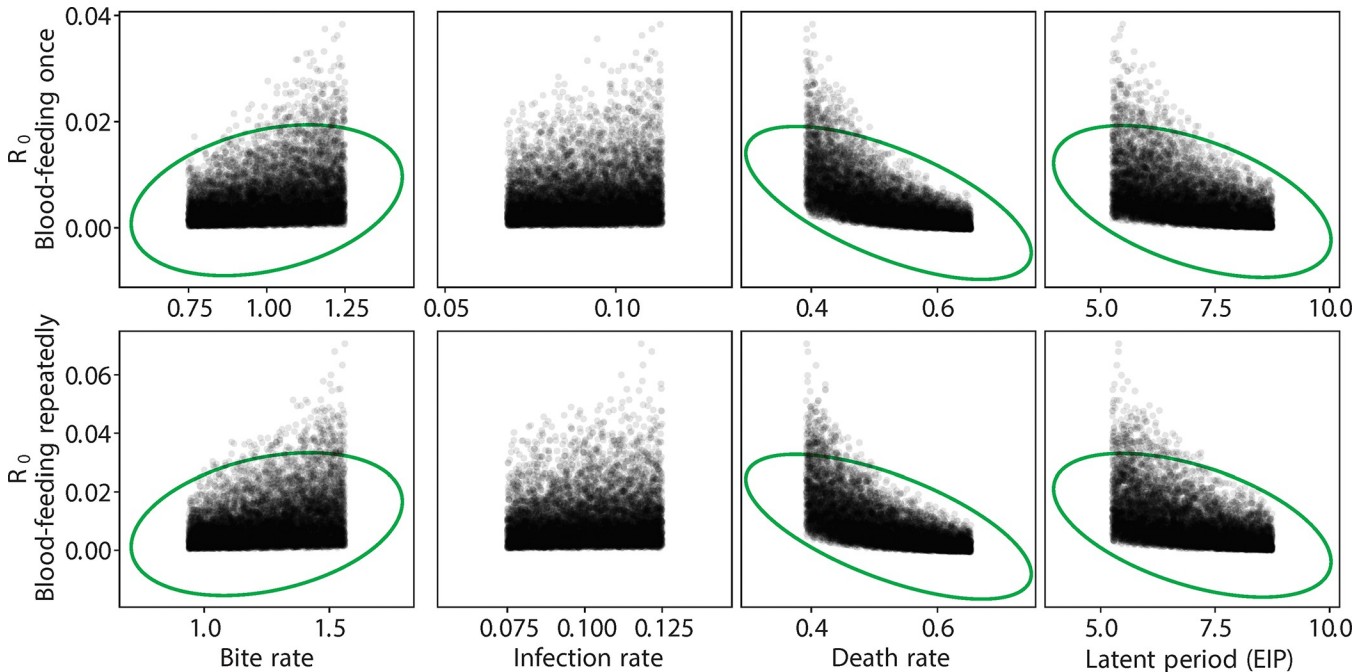

**Fig 6. $R_0$ values vs. control parameters for different blood-feeding experiments considering *Wolbachia*-infected fertile mosquitoes.** Green circles indicate the parameters contributing significantly.

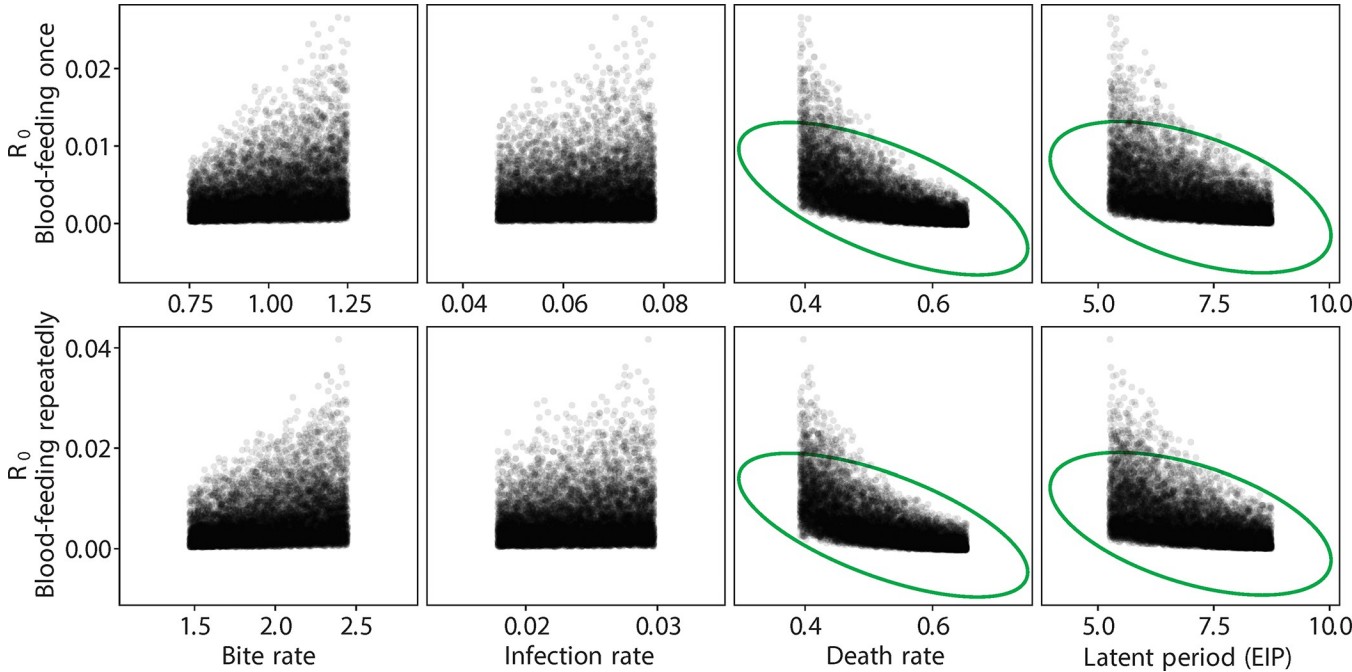

**Fig 7. $R_0$ values vs. control parameters for different blood-feeding experiments considering *Wolbachia*-infected infertile mosquitoes.** Green circles indicate the parameters contributing significantly.

## Discussion

In the last decade, *Wolbachia* was released into *Ae. aegypti* field populations in several countries around the world to inhibit the transmission of arboviral diseases [1,2]. It was recently discovered *Ae. aegypti* females infected with the *Wolbachia* *w*AlbB strain can lack matured ovaries, become infertile, and show higher feeding rates than fertile *Wolbachia*-infected or wildtype uninfected females after experiencing an extended period of egg quiescence [25,26]. The increased feeding rate poses a potential challenge to the efficacy of *Wolbachia* as a biocontrol agent for controlling arboviral diseases. Here we conducted measurements of longevity, probing frequency, and quantified dengue virus in mosquito legs following a single or multiple infectious blood meal with DENV. Additionally, we modeled the effect of these measured parameters on virus transmission.

Our data show that adults infected with *Wolbachia* exhibit a slight reduction in survival compared to wild type after storage. We know, however, from previously published works in *Ae. aegypti* [3,4] that *Wolbachia* infection, even without egg storage, can cause similar scale reductions in lifespan to what was measured here. The reduction in survival is nearly linear, however, which is unusual and warrants future comparisons with *Wolbachia*-infected mosquitoes without a history of egg storage. Our detailed time course of feeding behavior in this study demonstrates how the probing behavior of *Wolbachia*-infected mosquitoes (dominated by the infertile phenotype) is increased. Importantly, the differences relative to the wild type are disproportionately due to behavior during the early dpi. After a single feed, wild type mosquitoes are not interested in probing again until at least day 4. The *Wolbachia*-infected individuals begin feeding almost immediately, at a rate roughly 10-fold higher. Even after dpi 4, the *Wolbachia*-infected population continued to probe at roughly twice the rate. These differences in behavior equate to both an increase in viral loads and viral prevalence in *Wolbachia*-infected infertile over fertile mosquitoes through increased blood meal consumption.

Our DENV prevalence data were not as expected because they lacked evidence of *Wolbachia*-mediated virus blocking. In one replicate, the wild type line exhibited the lowest rates of infection and in another replicate, it did not differ from either of the *Wolbachia*-infected lines. The viral load data, in contrast, exhibited the predicted pattern of blocking, with higher loads in the wild type line compared to the *Wolbachia*-infected line. Previous studies on pathogen blocking in disseminated proxy tissues (heads, wings, legs) have shown reduced prevalence of both DENV [3] and ZIKV [39] well beyond 8 dpi in the presence of *w*AlbB in contrast to our data. The lack of blocking could result from incomplete homogenization of the Wolbachia uninfected and infected lines, although at three rounds of backcrossing they should be >80% similar. Alternatively, these data suggest that long-term egg storage may affect *Wolbachia's* blocking ability with respect to prevalence. Our viral load data, in contrast, did show strong evidence of blocking. We should note that the disconnect we see between prevalence and load viral load, while unusual [2], is not without precedent. A recent study from our group for Jamestown Canyon Virus indicated that these two traits can indeed be independent of one another [40]. In this latter case, *Wolbachia* was successful at preventing infection in individuals/tissues but was not effective at controlling viral load once infection was initiated. Here, we see the opposite, that infection of the tissues was very successful, but there was then subsequent moderation of viral loads.

Our modeling attempted to capture the additive effects of the increased biting rate due to *Wolbachia*-induced egg storage effects on transmission potential. We could not empirically measure the EIP in the critical early time points because we could not begin sampling until 8 dpi to allow for our repeated early feedings. We therefore used conservative estimates of EIP from the literature, with *Wolbachia*-infected lines exhibiting a later EIP. Repeat feeding, however, as seen in infertile females from stored eggs, might reduce the differential between wild-type and *Wolbachia*-infected fertile mosquitoes. As EIP is the dominant factor in transmission, given its power term in the Ross-MacDonald equation, any such reductions in EIP could substantially increase transmission. Additionally, other studies have shown that as viral loads rise, *Wolbachia*-mediated blocking is less effective [37]. Repeated feeding may reduce the efficacy of *Wolbachia* with greater exposure to virus. But while our modeling indicates *Wolbachia*-mediated blocking is more powerful than the infertility-induced repeat biting effects on load and prevalence, we do not think they have been fully captured in our design without a true empirical measure of EIP.

## Conclusions

This study has raised several questions that should be explored in future empirical work. First, an expanded set of tissues, including salivary glands, should be assessed for viral prevalence and load in *w*AlbB-infected mosquitoes from stored eggs versus unstored to determine if blocking strength is indeed altered. Similarly, adult survival of stored and unstored lines should be compared. Second, we suggest that EIP for DENV be measured in the context of egg storage and *Wolbachia w*AlbB infection to get a better estimate of the relative importance of infertility on transmission. Third, previous work has suggested that infertility in response to egg storage is greater for the *w*AlbB than the *w*Mel strain [25]. The effect of egg storage on virus transmissibility for *w*Mel-infected mosquitoes should be measured and compared to our data. In terms of broader impact, our study also raises issues that need to be considered in field releases where there are long dry seasons that may force egg quiescence. The releases being developed in Saudi Arabia may provide an interesting test of the importance of these effects in the field [41]. If that quiescence is long enough and is experienced by a high proportion of individuals in the population, *Wolbachia*-infected individuals may be at a disadvantage

reproductively, threatening spread. We found ~80% infertility after 11 weeks, but the previous study also found 25% at only 6 weeks [25]. Second, such storage may lead to greater biting in the resulting infertile females, which could increase transmission in the short term and reduce the efficacy of *Wolbachia*-mediated blocking. Last, any increased biting rates associated with *Wolbachia* infection may compromise community acceptance of its deployment. As the ideal scenario for successful *Wolbachia* replacement releases in the field includes both high female reproductive success and maximal reductions in virus transmission, further studies are needed to fully examine the relationships between storage length, infertility, and transmission and how they may affect the efficacy of *Wolbachia* over dryer landscapes.

## Supporting information

**S1 Fig. Viral infection following repeat feeding following 11 weeks of egg storage for Replicate 2.** (A) DENV-2 infection prevalence and (B) DENV-2 load in adult female *Aedes aegypti* legs at 8, 11, and 14 days post-infection (dpi). Females were provided with infectious blood every day from 1 to 7 dpi (post the initial infectious blood meal at 0 dpi). Each collection point x treatment is represented by 24 individuals. The fertility status of *Wolbachia*-infected females was identified through ovarian dissection.
(PDF)

**S2 Fig. *Wolbachia* loads in mosquito bodies post-storage and a single feeding event did not differ (p = 0.057) between fertile and infertile individuals.**
(PDF)

## Acknowledgments

In memory of Professor Howie Weiss (Biology Department, The Pennsylvania State University) for his support of ARV and his enthusiasm for modeling host:parasite interactions.

## Author Contributions

**Conceptualization:** Meng-Jia Lau, Ary A. Hoffmann, Elizabeth A. McGraw.

**Formal analysis:** Andrés R. Valdez.

**Funding acquisition:** Elizabeth A. McGraw.

**Investigation:** Meng-Jia Lau, Matthew J. Jones.

**Methodology:** Andrés R. Valdez.

**Project administration:** Elizabeth A. McGraw.

**Resources:** Ary A. Hoffmann.

**Supervision:** Igor Aranson.

**Writing – original draft:** Meng-Jia Lau, Andrés R. Valdez, Elizabeth A. McGraw.

**Writing – review & editing:** Meng-Jia Lau, Andrés R. Valdez, Matthew J. Jones, Igor Aranson, Ary A. Hoffmann, Elizabeth A. McGraw.

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
