## [Decision Letter · Decision Letter 0]

23 Apr 2024

Dear Prof McGraw,

Thank you very much for submitting your manuscript "The effect of egg quiescence in Wolbachia-infected Aedes aegypti on dengue virus transmission potential" for consideration at PLOS Neglected Tropical Diseases. As with all papers reviewed by the journal, your manuscript was reviewed by members of the editorial board and by several independent reviewers. In light of the reviews (below this email), we would like to invite the resubmission of a significantly-revised version that takes into account the reviewers' comments. 

We cannot make any decision about publication until we have seen the revised manuscript and your response to the reviewers' comments. Your revised manuscript is also likely to be sent to reviewers for further evaluation.

Sincerely,

Denis Voronin

Academic Editor

David Safronetz

Section Editor

Reviewer's Responses to Questions

**Key Review Criteria Required for Acceptance?**

**Methods**

-Are the objectives of the study clearly articulated with a clear testable hypothesis stated?

-Is the study design appropriate to address the stated objectives?

-Is the population clearly described and appropriate for the hypothesis being tested?

-Is the sample size sufficient to ensure adequate power to address the hypothesis being tested?

-Were correct statistical analysis used to support conclusions?

-Are there concerns about ethical or regulatory requirements being met?

Reviewer #1: (No Response)

Reviewer #2: The authors have used only samples obtained from quiescent eggs, so they cannot conclude that quiescence is a cause for those observed results.

Is there any reference about the mosquito colony used as control? If yes, please cite it in the mosquito-rearing section.

Both in the methodology section and in the results, they describe that they evaluated the fertility of mosquitoes with and without Wolbachia to compare. Therefore, it is important to make the purpose of this analysis clearer.

Was there any other tissue analyzed during the three dpi (8, 11 and 14), or were only the legs investigated? If only legs were analyzed, please replace Mosquito tissue samples with Mosquito legs. 

Explain why only for the Wolbachia infected mosquitoes the bodies were also collected.

In this section, the total number of groups investigated is not described properly. We think about four groups (1: Wolbachia infected fed once, 2: Wolbachia infected fed multiple times, 3: Wolbachia free fed once, and 4: Wolbachia free fed multiple times, however, when we see Figure 3 we observe six groups, so I suggest to explain this better in the methodology section.

Please add dpi (days post infection) somewhere in methodology

Reviewer #3: Yes, although there are some comments about the experimental design adopted.

**Results**

-Does the analysis presented match the analysis plan?

-Are the results clearly and completely presented?

-Are the figures (Tables, Images) of sufficient quality for clarity?

Reviewer #1: (No Response)

Reviewer #2: I think that results should be described better according to groups described in the methodology sections. For instance, in the introduction the authors mention "two data sets", but it is not clear what they are. Groups are not well defined and thus, results are very confusing.

Reviewer #3: Yes

**Conclusions**

-Are the conclusions supported by the data presented?

-Are the limitations of analysis clearly described?

-Do the authors discuss how these data can be helpful to advance our understanding of the topic under study?

-Is public health relevance addressed?

Reviewer #1: (No Response)

Reviewer #2: I do not agree with the first sentence in the conclusion section. Legs or other tissues such as head and midgut are fine to detect infection/prevalence rate. Salivary glands are normally used for estimating transmission rate, which is not the purpose of this study. Most of the conclusions are based on the failure of the methodological design, which seems to be the main weakness of the study. At the end, I think that the big question which is stated in the title could not be confirmed in this study, regarding the egg quiescence.

Reviewer #3: Yes

**Editorial and Data Presentation Modifications?**

Reviewer #1: (No Response)

Reviewer #2: Introduction

First sentence should be corrected Wolbachia is widespread in arthropods not only in insects. The way it is written seems to be specific to insects.

Some references were misused e.g. Reference 2 in the introduction section does not refers to Zika. Reference 27 is about malaria and not virus.

In the second paragraph, second line, the sentence seems to lack some words ("the expression of C ,)"

Discussion

In the first paragraph the authors mention "It was recently discovered Ae. aegyp- females infected with the Wolbachia wAlbB strain can lack matured ovaries, become inferOle, and show higher feeding rates than ferOle Wolbachiainfected or wildtype uninfected females aYer experiencing an extended period of egg diapause

[3, 4]" However these two references have no information about that. Please, be aware that Aedes aegypti eggs do not enter diapause, but quiescence. Correcte that sentence and cite proper references.

The last sentence in the first page of discussion section "wolbachia-infected infertile over infertile mosquitoes" is not clear.

The sentence "Our DENV prevalence data were not as expected because they lacked evidence of Wolbachia mediated

virus blocking. In one replicate, the wild type line exhibited the lowest rates of infection and in another replicate, it did not differ from either of the Wolbachia-infected lines....We propose two explanations for the prevalence data. The first is that the lack of blocking is the result of our choice of tissue (legs)." sounds very deterministic as if it will be necessary to change the methods to find the results the authors want to find. 

The authors could mention the inconvenience of increased bites from wolbachia-infected mosquitoes released into the field, which could compromise communities' acceptance of these types of strategies.

Reviewer #3: Added below.

**Summary and General Comments**

Reviewer #1: Lau et al carried out a follow up study on the effect of longer-term quiescence of Aedes aegypti eggs, which causes infertility and higher biting rates. The study, which utilised the wAlbB-infected mosquitoes, focused on their fitness, probing behaviours, and vector competence, considering multiple biting events. The results confirmed the previous finding that longer quiescence affects the mosquito’s fecundity, but importantly an increase in the transmission of dengue virus by the Wolbachia-infected infertile mosquitoes. The outcomes are significant for release strategies considering the climatic conditions of the area. The manuscript is written well, the conclusions are supported by the results, and it furthers the knowledge in the field of biological control using Wolbachia. Minor comments are listed below.

Minor comments

Note: please make it a habit to add continuous line numbers. Page numbers were not added either. So, below are based on the generated pdf.

Page 10 > life history traits: how was fertility status determined by examination of the ovaries? This becomes clearer later in the manuscript, but I suggest adding it to the methods section, where it first appears.

Page 11: in most places where “media” is mentioned, it should be “medium”. For example “RPMI 1640 media”.

Pages 12 and 13, top: q-PCR > qPCR

Page 13 > statistical analysis: Gaussian distribution

Page 15, para 2: join the two sentences “When we provided ……, infertile females….”

Page 15, last para, line 2: is Fig. 4 meant to be Fig. 3?

Reviewer #2: My main concern is that the authors claim in the last paragraph of the introduction that they examine the effect of egg quiescence and multiple blood meals on DENV transmissibility in Wolbachia-infected mosquitos. However, in all the experiments only samples from quiescent eggs were used. So, there was no comparison between quiescent and non-quiescent eggs to conclude that quiescence caused all the observed changes. 

In addition, in the methodology section, how the groups of analyzed samples are described is very confusing and needs to be better explained. 

Why didn´t the authors check for the presence of Wolbachia in the Wolbachia-free group. It is known that Wolbachia is invasive and can colonize Wolbachia free samples in the Lab. How can we make sure that they are not positive for Wolbachia?

Reviewer #3: The manuscript entitled ‘The effect of egg quiescence in Wolbachia-infected Aedes aegypti on dengue virus transmission potential’ deals with an intriguing question regarding the use of Wolbachia to reduce dengue transmission. This endosymbiont has been deployed in several countries with promising results based on case counting reports. There is strong evidence showing reduction in dengue transmission in areas where Wolbachia has been deployed. And this effect has been shown for both strains used in the field: wMel and wAlbB. This manuscript reports some life-history traits of wAlbB-infected mosquitoes originated from quiescent eggs stored for 11 weeks under specific laboratory conditions. The manuscript lines are not numbered, making it harder for both reviewer and authors to locate in the text specific typos, for instance.

Major comment:

1. Authors adopted an experimental design in which they decided to screen DENV-2 in mosquito legs rather than in mosquito saliva. Although they recognize saliva as the golden standard for transmission dynamics, it is still unclear the reason why this design was chosen. 

Minor comments:

1. Methods. Traditional backcrossing would expect mating Wolbachia-infected females with wild uninfected males for around 5-6 generations. By doing so, we should expect around 80% of Mexican genetics rather than a >95%. Why not doing this backcrossing for a few more generations? Please discuss whether this incomplete backcrossing could influence the results.

2. Methods. It was not clear to me the reason why eggs were storage for 11 weeks, not less, not more. Such research question would be better answered if a range of storage times are explored, not only a single timepoint at 11 weeks.

3. Methods. By stating ‘to check their fertility status by examining their ovaries to obtain the proportion of infertile females’ it would be didactic to add a reference with a picture or adding your own picture as a supplementary file.

4. Methods, Life history traits subsection, 2nd paragraph. Please add that this blood meal was composed of DENV-uninfected blood.

5. Methods. How to record all probing events in a cage of 30-40 individuals? Count those blood fed is ok but counting the number of probes in 30-40 starved individuals simultaneously is hard. Can you provide more details on how this data was obtained, e.g., how often it was needed to go back to video record?

6. Methods. Please provide more information about the DENV-2 isolate used. It seems it was isolated in Australia, but mosquitoes have a 80% Mexican background. Do authors see it as a potential source of complexity to analyze their results?

7. Results. Something is missing in the link between this sentence ‘When we provided pairwise comparisons for Wolbachia-infected fertile and infertile females’ with the others.

8. Please standardize the use of dpi or DPI.

9. Discussion. The Wolbachia-infected mosquitoes probe at twice the rate of the Wolbachia-uninfected mosquitoes, even after 4 dpi. But it also reflected on a higher amount of ingested blood?

10. Results. In Figure 3A, y-axis, please specify it is DENV-2 infection prevalence.

PLOS authors have the option to publish the peer review history of their article (what does this mean?). If published, this will include your full peer review and any attached files.

Reviewer #1: No

Reviewer #2: No

Reviewer #3: No

Figure Files:

Data Requirements:

Reproducibility:

To enhance the reproducibility of your results, we recommend that you deposit your laboratory protocols in protocols.io, where a protocol can be assigned its own identifier (DOI) such that it can be cited independently in the future. Additionally, PLOS ONE offers an option to publish peer-reviewed clinical study protocols. Read more information

---

## [Decision Letter · Decision Letter 1]

21 Jun 2024

Dear Prof McGraw,

We are pleased to inform you that your manuscript 'The effect of repeat feeding on dengue virus transmission potential in Wolbachia-infected Aedes aegypti following extended egg quiescence' has been provisionally accepted for publication in PLOS Neglected Tropical Diseases.

Best regards,

Denis Voronin

Academic Editor

David Safronetz

Section Editor

Reviewer's Responses to Questions

**Key Review Criteria Required for Acceptance?**

**Methods**

-Are the objectives of the study clearly articulated with a clear testable hypothesis stated?

-Is the study design appropriate to address the stated objectives?

-Is the population clearly described and appropriate for the hypothesis being tested?

-Is the sample size sufficient to ensure adequate power to address the hypothesis being tested?

-Were correct statistical analysis used to support conclusions?

-Are there concerns about ethical or regulatory requirements being met?

Reviewer #3: (No Response)

**Results**

-Does the analysis presented match the analysis plan?

-Are the results clearly and completely presented?

-Are the figures (Tables, Images) of sufficient quality for clarity?

Reviewer #3: (No Response)

**Conclusions**

-Are the conclusions supported by the data presented?

-Are the limitations of analysis clearly described?

-Do the authors discuss how these data can be helpful to advance our understanding of the topic under study?

-Is public health relevance addressed?

Reviewer #3: (No Response)

**Editorial and Data Presentation Modifications?**

Reviewer #3: (No Response)

**Summary and General Comments**

Reviewer #3: (No Response)

PLOS authors have the option to publish the peer review history of their article (what does this mean?). If published, this will include your full peer review and any attached files.

Reviewer #3: No

---

## [Editor Report · Acceptance letter]

2 Jul 2024

Dear Prof McGraw,

We are delighted to inform you that your manuscript, "The effect of repeat feeding on dengue virus transmission potential in Wolbachia-infected Aedes aegypti following extended egg quiescence," has been formally accepted for publication in PLOS Neglected Tropical Diseases.

Best regards,

Shaden Kamhawi

co-Editor-in-Chief

Paul Brindley

co-Editor-in-Chief
